# MG-NeRF: Multimodal Representation Learning for Generalizable NeRF

## Abstract

Neural Radiance Field (NeRF) is a novel view synthesis model for three-dimensional space implicit representation, which has swept the field of computer vision. However, vanilla NeRF overfits to a single scene and fails to synthesize novel views of unseen scenes. It is training expensive to learn per-scene representation so that generalization capability of NeRF has aroused tremendous attention. Previous works address the challenge through multi-view learning methods and achieve remarkable results. They convert novel view synthesis to multi-view image-based interpolation problem. These methods focus on image modality merely, while overlooking other meaningful multimodal knowledge. We propose MG-NeRF, a novel learning framework that incorporates multimodal information to polish the generalization performance of NeRF. To our best knowledge, we are the first to apply multimodal to NeRF's generalization. We employ a vision language pre-training (VLP) framework to generate text information for each scene. Then, image and text features are fused and fed to NeRF. Due to the alignment of image and text modalities, we bring in a semantic loss to encourage NeRF to synthesize reasonable novel views. For positional encoding, a frequency regularization mechanism is introduced to prevent NeRF from overfitting to high frequency information. We show that MG-NeRF achieves appreciable achievement on novel view synthesis of unseen scenes even trained with considerably less resources than prior work. We will public our code once upon acceptance.

## 1 Introduction

Synthesizing photorealistic images has been a prominent subject of research in the fields of computer vision and computer graphics, driven by the growing demand for virtual reality/augmented reality (VR/AR) experiences. Traditionally, rendering algorithms such as rasterization and ray tracing have been employed to generate high-quality synthetic images of a scene. However, these methods require explicit input of all physical parameters associated with the scene, including camera settings, object materials, and illumination conditions. Estimating these properties from existing observations, known as inverse rendering, poses significant challenges when aiming to generate controlled images that faithfully represent real-world scenes.

In contrast, neural rendering has emerged as a rapidly advancing field that enables compact representations of scenes and leverages neural networks to learn rendering from existing observations. Similar to traditional computer graphics, the primary objective of neural rendering is to generate controlled, photorealistic images. This includes tasks such as new viewpoint compositing, relighting, scene morphing, and compositing. One particularly popular example in recent years is the NeRF (Mildenhall et al. (2020)). NeRF aims to decouple the modeling and rendering processes by solely learning the 3D scene representation and relying on established rendering functions from computer graphics for supervision. This approach has found applications in various domains, including robotics, urban mapping, and autonomous navigation.

NeRF operates by capturing multiple images of a scene from known viewpoints, along with corresponding internal and external parameters for each image. Unlike traditional methods, NeRF bypasses the intermediate 3D reconstruction process and directly synthesizes images from novel views using only the positional information and the captured images. Under the NeRF-based representation, the 3D space is represented as a collection of continuous and learnable radiation fields.

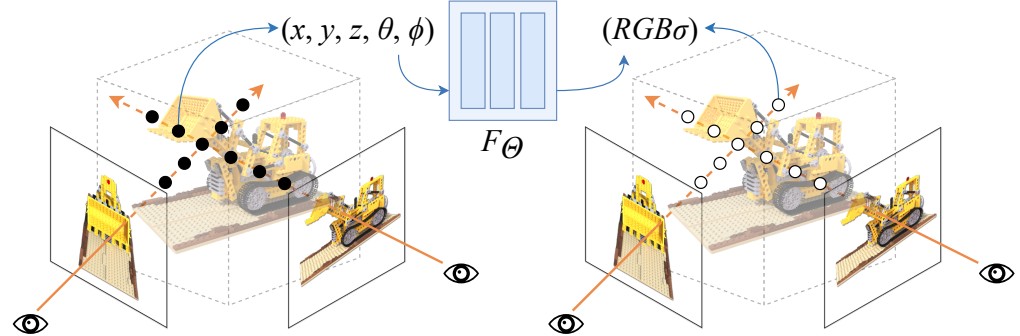

Figure 1: Vanilla NeRF architecture. Vanilla NeRF approximate this continuous 5D scene representation with an MLP network $F_\Theta : (x, d) \to (c, \sigma)$ and optimize its weights $\Theta$ to map from each input 5D coordinate to its corresponding volume density and directional emitted color, where $x = (x, y, z)$ indicates location, $d = (\theta, \phi)$ indicates view direction, $c = (R, G, B)$ indicates color, $\sigma$ indicates volume density.

These fields are learned from the input viewpoints and positional information to determine density and color attributes.

However, NeRF's reliance on per-scene optimization renders it susceptible to overfitting, limiting its ability to generate novel views in unseen scenes. This inherent limitation hinders the widespread application of NeRF. Consequently, there is a pressing need for a more generalizable NeRF approach. To address this challenge, researchers have explored various methods and techniques. MVSNeRF (Chen et al. (2021)) pretrains on multi-view datasets and incorporate per-scene optimization at test time. pixelNeRF (Yu et al. (2021b)) and IBRNet (Wang et al. (2021)) condition NeRF on a set of source views by extracting scene features from those source views. NeuRay (Liu et al. (2022)) proposes a visibility computational mechanism that can quickly and effectively help infer occlusion relations and improve rendering quality. GPNR (Suhail et al. (2022)) addresses the generalizability problem of synthesizing unseen scenes by using a Transformer sequence with a 4D light field for representation normalized positional encoding. GNT (Varma et al. (2022)) proposes an attention-based generalizable NeRF Transformer architecture. These existing methods are based on the image modality.

In this article, we present a novel approach, MG-NeRF, aimed at enhancing the generalization capabilities of NeRF by leveraging multimodal knowledge. Our contributions are significant as we are the first to incorporate multimodal information into NeRF's generalization process. To achieve robust generalization across diverse scenes and multimodal data, we employ IMAGEBIND (Girdhar et al. (2023)), a state-of-the-art zero-shot multimodal-aligned pretrained model, as the encoder in our framework. Additionally, we introduce LoRA (Hu et al. (2021)), a technique for fine-tuning IMAGEBIND (Girdhar et al. (2023)), to further enhance its performance. To refine NeRF, we introduce semantic loss and frequency regularization, which play a crucial role in improving its accuracy and fidelity. The proposed MG-NeRF framework is summarized in Fig. 2. Experimental results demonstrate that MG-NeRF achieves remarkable performance and significant advancements in the field. Our key contributions include:

1. The first multimodal learning framework that improves NeRF's generalization ability.
2. Generated Text captions for each scene to get text modality information.
3. A semantic loss to encourage realistic rendering.
4. A frequency regularization mechanism to relieve artifacts.

## 2 RELATE WORK

**NeRF** Vanilla NeRF represents a scene as a continues 5D function $F_\Theta : (x, d) \to (c, \sigma)$ that outputs color $c = RGB$ and volume density $\sigma$ in each direction $d = (\theta, \phi)$ at each point $x = (x, y, z)$. It utilizes Multi-Layer Perception (MLP) to approximate the function, as Fig. 1 shows. Generally, the training process of NeRF is self-supervised. It calculates Mean-Squared Loss (MSE) between

Ground Truth $G$ views and synthetic views $S$, which is called reconstruction loss $L_{rec}$, to optimize the weight $\Theta$, seen in Eq. 1. In order to make the representation more multi-view consistent, vanilla NeRF limits volume density $\sigma$ to location $\boldsymbol{x}$, while limits color to location $\boldsymbol{x}$ and view direction $\boldsymbol{d}$.

$$L_{rec} = ||S - G||^2 \tag{1}$$

Many branches of researches on NeRF have been developing prosperously. Mip-NeRF series (Barron et al. (2021); Verbin et al. (2022); Mildenhall et al. (2022)), CodeNeRF (Jang & Agapito (2021)) and PointNeRF (Xu et al. (2022)) study the fundamentals of NeRF. Instant-NGP (Müller et al. (2022)), Plenoctree (Yu et al. (2021a)) and TensorRF (Chen et al. (2022)) speed up NeRF. GIRAFFE (Niemeyer & Geiger (2021)), Edit-NeRF (Liu et al. (2021)) and CLIP-NeRF (Wang et al. (2022)) realize conditional NeRF and manipulation. NeRF-W (Martin-Brualla et al. (2021)), NeRF++ (Zhang et al. (2020)) and GIRAFFE (Niemeyer & Geiger (2021)) decompose NeRF into foreground and background parts. iMAP (Sucar et al. (2021)), NICE-SLAM (Zhu et al. (2022)) and SCNeRF (Jeong et al. (2021)) focus on pose estimation.

**Pretrained Large Model** A pretrained large model refers to a neural network model that has been trained on extensive datasets, typically containing billions of parameters. This model exhibits strong performance across a wide range of tasks. Through the training process, it acquires valuable features from the vast amount of data, which can be effectively transferred to specific tasks. As a result, it demonstrates excellent performance on multiple tasks without requiring extensive task-specific retraining. The concept of pretrained large models has garnered significant attention in the field of artificial intelligence and has already yielded remarkable outcomes in domains such as natural language processing, computer vision, and speech recognition. BERT (Devlin et al. (2018)), GPT (Brown et al. (2020)) and ViT (Dosovitskiy et al. (2020)) are three classical single-modality pretrained large models, while multimodal pretrained large models come out one after another. CLIP (Radford et al. (2021)) adopts the idea of comparative learning and receives two different modalities of data for training at the same time, and performs well in tasks such as text-image retrieval, image classification, and text-based image generation. IMAGEBIND (Girdhar et al. (2023)) utilizes Image Bind to learn an embedding space that contains information about all modalities. It achieves alignment among six modalities by aligning the embedding of each modality with the image embedding, even if they are not observed simultaneously with each other.

The utilization of large models in the context of NeRF is gaining traction. CLIP-NeRF (Wang et al. (2022)) employs CLIP (Radford et al. (2021)) as image/text encoder to extract the corresponding feature embedding for the shape and appearance mappers to lean a local step in the latent space for shape and appearance manipulation, respectively. FeatureNeRF (Ye et al. (2023)) trains NeRF by distilling pre-trained vision foundation models such as DINO (Caron et al. (2021)) and Latent Diffusion (Rombach et al. (2022)).

## 3 METHOD

The schematic overview of MG-NeRF is depicted in Fig. 2, illustrating its five key components: data preparation, encoder, loss, frequency regularization, and the NeRF Network. In the data preparation stage, we employ Tag2Text (Huang et al. (2023)) to extend image data into text data and, if necessary, obtain pose information using COLMAP (Schönberger & Frahm (2016); Schönberger et al. (2016)). The encoder module leverages the power of IMAGEBIND (Girdhar et al. (2023)) and LoRA (Hu et al. (2021)) to extract aligned multimodal features, which are then fused before being passed to the NeRF network. To address high-frequency information, we introduce a mask mechanism as part of the frequency regularization process. In addition to the classical rendering loss mentioned in Sec. 2, we propose a semantic loss to encourage more plausible rendering outcomes. Finally, the NeRF network adopts a two-stage transformer architecture, similar to GNT (Varma et al. (2022)), to effectively model the scene.

### 3.1 TEXT CAPTION GENERATION

The currently available neural radiation field datasets lack text modalities, requiring manual addition of text descriptions. This can be done through two options: (1) manual labeling, and (2) automated labeling. As discussed earlier, it is crucial for scene descriptions to exhibit semantic consistency, meaning that the text description should remain the same regardless of the viewing angle. Given that

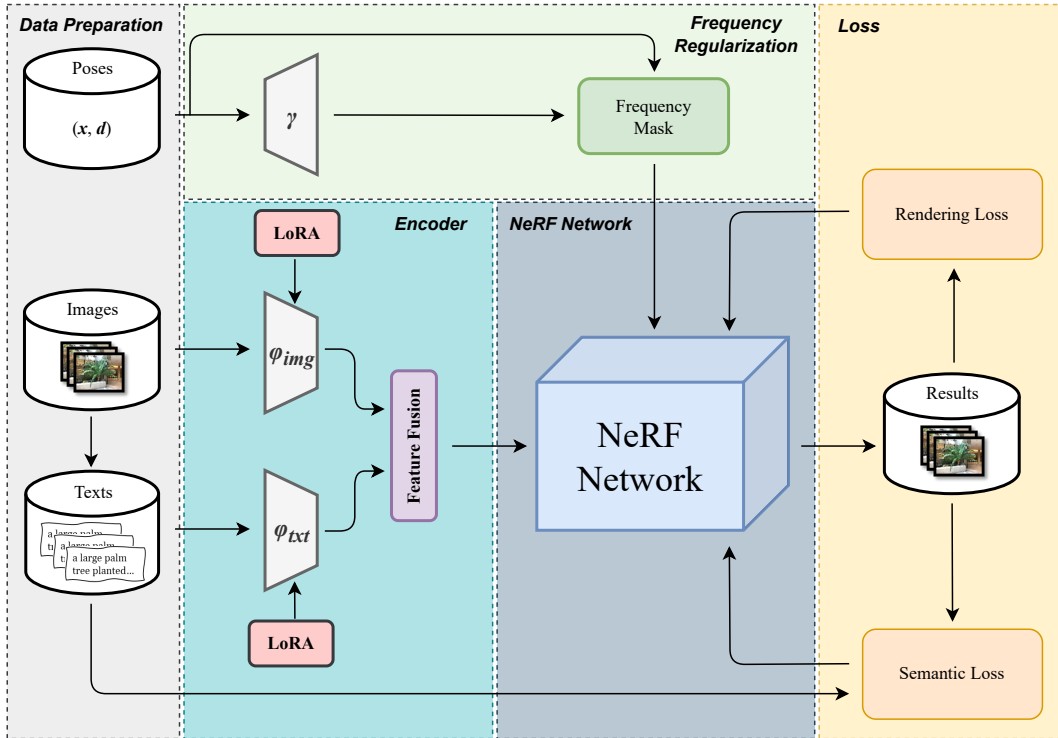

Figure 2: The framework of the proposed MG-NeRF. A key contribution of our work is the extension of single-modal learning (specifically, image modality) to multi-modal learning, incorporating both image and text modalities. Textual data is generated for each scene, and to effectively integrate the information from different modalities, we employ a pretrained model that can generate a joint embedding across these modalities, as opposed to using a conventional ResNet architecture. Furthermore, we leverage the LoRA (Hu et al. (2021)) technique to efficiently fine-tune this pretrained model. The resulting embeddings are then fused to enhance the feature representation. To address the positional information, we employ a frequency regularization method that prevents NeRF from overfitting to high-frequency components. Additionally, we introduce two types of loss functions that work together during the optimization stage, facilitating improved learning of NeRF.

a scene often consists of tens or hundreds of images, manually annotating all the mainstream datasets such as LLFF dataset (Mildenhall et al. (2019)), NeRF Synthetic dataset (Mildenhall et al. (2020)), IBRNet Collected dataset, Spaces dataset (Flynn et al. (2019)), Google Scanned Objects dataset (Downs et al. (2022)), RealEstate10K dataset (Zhou et al. (2018)), etc., which collectively contain thousands of scenes and tens of thousands of images (approximately 827,284 images), would require significant manpower and time. Moreover, different annotators may introduce varying degrees of bias. Therefore, the use of pretrained large models is considered to address this task.

In this project, we propose the implementation of text description generation using the pretrained visual-linguistic macromodel Tag2Text (Huang et al. (2023)). Tag2Text (Huang et al. (2023)) is an efficient and controllable visual-linguistic model that generates text descriptions guided by tags. It excels in recognizing tags for images across 3429 commonly used human-defined categories without the need for manual annotations. The efficient tag guidance significantly enhances the performance of the visual-linguistic model in both generation-based and alignment-based tasks. Moreover, Tag2Text (Huang et al. (2023)) offers controllability, allowing users to input desired tags and flexibly combine corresponding text based on the provided tags. For instance, when an image is uploaded, Tag2Text (Huang et al. (2023)) can recognize the objects within it and assign appropriate tags, generating text descriptions guided by these tags. Alternatively, users can specify the tags themselves. According to the semantic consistency of a single scene, we adopt **one-scene-one-description** principle. Specifically, a description is applicable to all the images from the same scene, for "a bull-

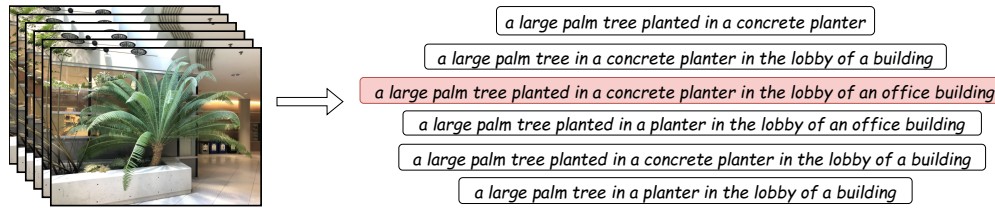

Figure 3: Text caption generation. For example, given several images from fern on the left (Milden-hall et al. (2019)), we utilize the pretrained model to generate captions of each image and then get six slightly different descriptions on the right. Next, we choose the longest one *"a large palm tree planted in a concrete planter in the lobby of an office building"* (in the red box) to describe the fern scene.

dozer is a bulldozer from any perspective" (Jain et al. (2021)). Suppose $N$ images in a scene are given, denoted as $\{I_i\}_{i=1}^{i=N}$, we feed each image into Tag2Text model $F$ and get $N$ text descriptions $\{F(I_i)\}_{i=1}^{i=N}$. Then, we calculate the length of each text description and select the longest one as the unified text description of the scene, as Fig.3 shows.

## 3.2 ENCODER

Previous approaches have primarily relied on classic ResNet-based or transformer-based encoders. In contrast, our work takes a different approach by leveraging the capabilities of IMAGEBIND (Girdhar et al. (2023)) to harness multimodal information. Among the various large-scale multimodal pretrained models available, IMAGEBIND (Girdhar et al. (2023)) stands out as it incorporates four additional modes of data: large-scale image-text pairs, self-supervised data, audio, depth, thermal, and inertial measurement unit (IMU) readings. Notably, IMAGEBIND (Girdhar et al. (2023)) has demonstrated superior performance in zero-shot classification and retrieval tasks across each of these modalities. Importantly, IMAGEBIND (Girdhar et al. (2023)) pioneers the integration of IMU, depth, and heat modalities in cross-modal domain retrieval tasks. This expanded range of modalities sets IMAGEBIND (Girdhar et al. (2023)) apart from models like CLIP (Zhou et al. (2022)) and ONE-PEACE (Wang et al. (2023)), offering greater flexibility and making it a valuable asset for our research.

In general, pretrained models often exhibit suboptimal performance when applied to new scenes or tasks. To address this limitation, we introduce LoRA (Hu et al. (2021)), a novel PEFT (Parameter-Efficient Fine-Tuning) method that facilitates efficient adaptation of pretrained large models to diverse downstream applications without the need to fine-tune all parameters of the model. Specifically, we extract image features and text features separately from the pretrained model and combine them within an adapter-like structure to obtain enriched features, as illustrated in Fig. 4. These enriched features are then fed into the NeRF network for further processing and synthesis.

## 3.3 LOSS

Our loss consists of two parts. The first is typical reconstruction loss $L_{rec}$, which calculates the L2 norm between ground truth and generated images, seen in Eq. 1. The second is semantic loss $L_{sec}$, which meters the semantic similarity between the scene and corresponding images. We utilize text captions $T$ presented in Sec. 3.1 to represent scenes' semantic information. Since IMAGEBIND (Girdhar et al. (2023)) is multimodal aligned, we reuse its image modality encoder $\phi_{img}$ to extract generated novel views' feature and text modality encoder $\phi_{txt}$ to extract scenes' feature. Then, these different modalities' features are encoded in a joint embedding space. After that, the semantic loss $L_{sec}$ is computed through cosine similarity, seen in Eq. 2.

$$L_{sec} = cos(\phi_{img}(S), \ \phi_{txt}(T)) \qquad (2)$$

Where $S$ and $T$ indicate novel views and text captions. At last, the total loss $L_{tot}$ is weighted sum of $L_{rec}$ and $L_{sec}$ with factors $\lambda$, $\mu$, seen in Eq. 3. In practice, $\lambda$ is set to 0.9 and $\mu$ is set to 0.1.

$$L_{tot} = \lambda L_{rec} + \mu L_{sec} \qquad (3)$$

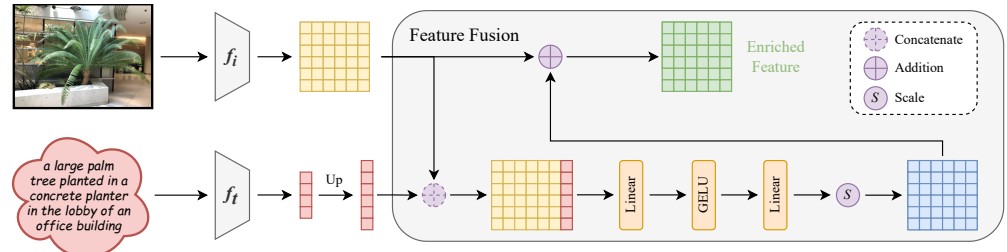

Figure 4: Feature fusion pipeline. After data preparation, images and text captions are well-paired when loading data. Images and text captions are input encoders $f_i$, $f_t$ to get original features. Then, linearly map text feature to higher dimension that aligns to image feature's size. We put forward an adapter-like structure (gray round rectangle) (Houlsby et al. (2019)) to perform feature fusion. Processed features are concatenated and are exerted through downwards linear mapping, GELU activation, upwards linear mapping, and scaling, where the scale factor $S$ is set to 0.1 in our experiments. At last, enriched feature (green blocks) is obtained by adding the output (blue blocks) and original image feature (yellow blocks).

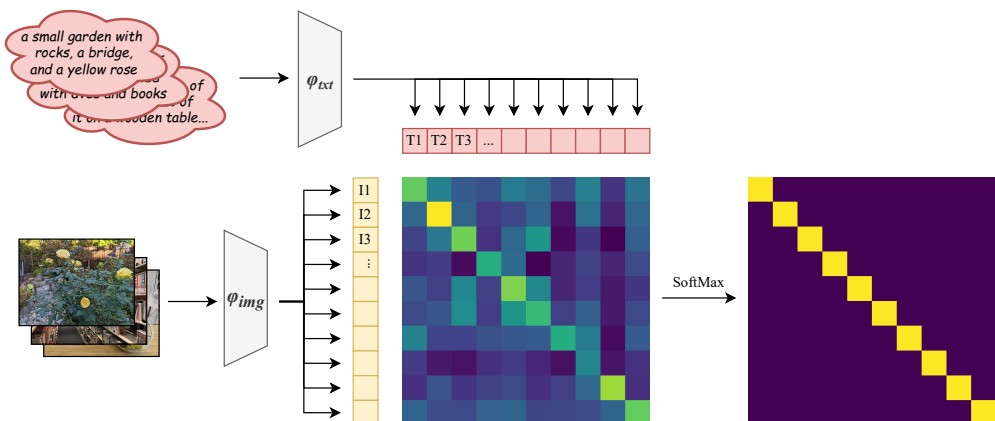

Figure 5: Semantic similarity between images and text captions. To verify that generated text captions are aligned with images, we randomly select several images with related text captions across different scenes of different datasets used in this article to compute the similarity in the manner of (Radford et al. (2021)). Image set $\{i_k\}_{k=1}^{k=N}$ is corresponding to text caption set $\{t_l\}_{l=1}^{l=N}$ if $k$ equals to $l$. After $\phi$ encoding, we get Images' embedding set $\{I_k\}_{k=1}^{k=N}$ and text captions' embedding set $\{T_l\}_{l=1}^{l=N}$. We arrange each image embedding in a vector $\boldsymbol{S} = (I_1, I_2, ..., I_N)$ (shown in yellow block), and text caption vector $\boldsymbol{T} = (T_1, T_2, ..., T_N)$ in the same way (shown in red block). $\boldsymbol{T}$ is multiplied by $\boldsymbol{S}^T$ on the left to get the distance confusion matrix $\boldsymbol{C} = \boldsymbol{S}^T\boldsymbol{T}$. The confusion matrix is plot in the middle. Brighter color indicated more similar. After Softmax operation, we get an identity matrix $\boldsymbol{I}_N$ (bright yellow indicates 1 and dark purple indicates 0) that proofs generated text captions are well aligned with images.

## 3.4 FREQUENCY REGULARIZATION

Since neural networks are biased towards learning low-frequency functions (Rahaman et al. (2019)), resulting in the disability of NeRF to synthesize high quality about the parts of the image that have drastic color and geometric variations (the high-frequency parts of the image). Researchers address the problem by introducing a high-frequency function, $\gamma(\cdot)$, which maps the inputs to a high-dimensional space, as Eq. 4 shows, where $\boldsymbol{p}$ refers to a vector. Concretely, $\boldsymbol{p}$ could be the location $\boldsymbol{x}$ or view direction $\boldsymbol{d}$. For $\boldsymbol{p} = \boldsymbol{x}$, the original 3D input is mapped to $3L$ space. In practice, $L$ is set to 10 for $\gamma(\boldsymbol{x})$ and 4 for $\gamma(\boldsymbol{d})$.

$$\gamma(\mathbf{p}) = \left[ sin(2^0\pi\mathbf{p}), cos(2^0\pi\mathbf{p}), ..., cos(2^{L-1}\pi\mathbf{p}) \right] \qquad (4)$$

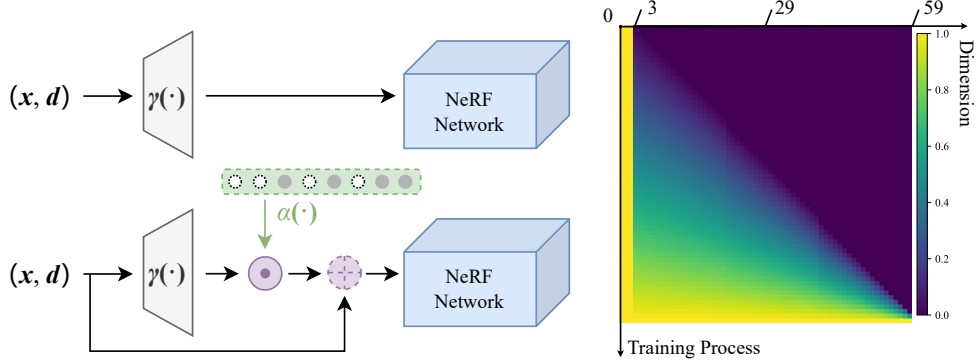

Figure 6: Frequency regularization. The top-left subfigure shows that position information is input to nerf directly after position encoding in accordance with Eq. 4. We bring in a novel frequency regularization mechanism after position encoding, marked as green part in the bottom-left subfigure. The frequency masks $\alpha$ exert on the encoded component to limit high frequency information Then, the result is concatenated with origin input, which is fed to the NeRF Network. The frequency masks is acquired by Eq. 5. The Visualization of whole frequency mask is plot in the right subfigure. As is shown, our frequency mask value is belong to [0, 1], where "0" means totally invisible and "1" implies not restriction. With training going, the limitation on high dimensions (3rd 59th) is gradually eased, while the first three dimensions (i.e, the original input) are visible all the time.

The aforementioned high-dimensional mapping methods are suitable for training NeRF using tens or hundreds of images. However, for a generalizable NeRF that aims to synthesize new viewpoints, typically only a few reference images are provided, resembling a few-shot neural rendering scenario. In such cases, the high-dimensional mapping approach can lead to overfitting of NeRF to high-frequency information. Research has shown that high-frequency mapping accelerates the convergence of the high-frequency components (Tancik et al. (2020)), which in turn results in the NeRF overfitting to high-frequency information and losing some low-frequency information. Inspired by (Yang et al. (2023)), we propose a frequency regularization mechanism that controls the visible frequency range of the NeRF using masks during the training process. This gradually opens up the frequency range from low to high frequencies, preventing the NeRF from overfitting to either low-frequency or high-frequency information.

Different from (Yang et al. (2023)), we linearly gradually release high frequency component to the NeRF. To begin with, for location $\mathbf{x} = (x, y, z)$, map it to $\gamma(\mathbf{x})$ according to Eq. 4. Second, concatenate $\gamma(\mathbf{x})$ with $\mathbf{x} = (x, y, z)$ to get $\mathbf{x}' = [\mathbf{x}, \gamma(\mathbf{x})]$ that possesses both low and high frequency information. Next, compute the frequency mask $\alpha$. Given the length of $\mathbf{x}'$ as $p$ and total training steps as $r$, the mask value $\alpha(u, v)$ of $u$-th dimension at $v$-th training step can be described as Eq. 5.

$$\alpha(u,v) = \begin{cases} 1 & , \ u < 3 \\ 0 & , \ u \geq 3, \ r(u-3) - pv < 0 \\ \dfrac{pv - r(u-3)}{pr - r(u-3)} & , \ else \end{cases} \tag{5}$$

where $u \in \{0, 1, ..., p\}$, $v \in \{0, 1, ..., r\}$. Consequently, we obtain the entire frequency mask $\alpha$, seen in Fig. 6. Finally, the location input at the $i$-th training step is $\mathbf{x}' \odot \alpha[i]$, where $\odot$ indicates Hadamard Product (Element-wise Multiplication) (Million (2007)).

## 4 EXPERIMENTS

### 4.1 IMPLEMENTATION DETAILS

**Datasets.** Following prior work (Varma et al. (2022)), we train our MG-NeRF on Real Icon Noface dataset, Spaces dataset (Flynn et al. (2019)), IBRNet Collected dataset, IBRNet Collected dataset, RealEstate10K dataset (Zhou et al. (2018)), Google Scanned Objects dataset (Downs et al. (2022)),

Table 1: Quantitative comparison on LLFF and NeRF Synthetic

| Models | Local Light Field Fusion (LLFF) | | | | NeRF Synthetic | | | |
|---|---|---|---|---|---|---|---|---|
| | PSNR↑ | SSIM↑ | LPIPS↓ | Avg↓ | PSNR↑ | SSIM↑ | LPIPS↓ | Avg↓ |
| pixelNeRF | 18.66 | 0.588 | 0.463 | 0.159 | 22.65 | 0.808 | 0.202 | 0.078 |
| MVSNeRF | 21.18 | 0.691 | 0.301 | 0.108 | 25.15 | 0.853 | 0.159 | 0.057 |
| IBRNet | 25.17 | 0.813 | 0.200 | 0.064 | 26.73 | 0.908 | 0.101 | 0.040 |
| NeuRay | 25.35 | 0.818 | 0.198 | 0.062 | 28.29 | 0.927 | 0.080 | 0.032 |
| GPNR | 25.72 | 0.880 | 0.175 | 0.055 | 26.48 | 0.944 | 0.091 | 0.036 |
| GNT | 25.86 | 0.867 | 0.116 | 0.047 | 27.29 | 0.937 | 0.056 | 0.029 |
| MG-NeRF | 24.28 | 0.810 | 0.176 | 0.066 | 24.86 | 0.910 | 0.089 | 0.044 |

Table 2: Ablation study on LLFF and NeRF Synthetic

| Models | Local Light Field Fusion (LLFF) | | | | NeRF Synthetic | | | |
|---|---|---|---|---|---|---|---|---|
| | PSNR↑ | SSIM↑ | LPIPS↓ | Avg↓ | PSNR↑ | SSIM↑ | LPIPS↓ | Avg↓ |
| w/o all | 22.91 | 0.766 | 0.212 | 0.081 | 18.66 | 0.588 | 0.463 | 0.159 |
| w/ Fusion | 23.69 | 0.792 | 0.192 | 0.072 | 23.71 | 0.893 | 0.100 | 0.052 |
| w/ LoRA | 23.55 | 0.788 | 0.196 | 0.074 | 21.17 | 0.868 | 0.128 | 0.071 |
| w/ Mask | 23.90 | 0.800 | 0.184 | 0.069 | 24.62 | 0.902 | 0.095 | 0.047 |
| w/ $L_{sec}$ | 23.75 | 0.795 | 0.190 | 0.071 | 22.12 | 0.877 | 0.122 | 0.064 |
| MG-NeRF | 24.28 | 0.810 | 0.176 | 0.066 | 24.86 | 0.910 | 0.089 | 0.044 |

and evaluate on LLFF dataset (Mildenhall et al. (2019)), NeRF Synthetic dataset (Mildenhall et al. (2020)). The ratios of training datasets are 0.3, 0.15, 0.35, 0.15, 0.05, respectively.

**Metrics.** We report PSNR (peak signal-to-noise ratio), SSIM (structural similarity index measure) (Wang et al. (2004)) and LPIPS (learned perceptual image patch similarity) (Zhang et al. (2018)) scores as quantitative results. We also report the geometric mean of MSE = $10^{-PSNR/10}$, $\sqrt{1 - SSIM}$ and LPIPS, following (Niemeyer et al. (2022)).

**Training / Inference Details.** We employ a training strategy where we freeze the IMAGEBIND (Girdhar et al. (2023)) and train NeRF Network with LoRA (Hu et al. (2021)). In all our experiments, we train for 250,000 steps with 512 rays sampled in each iteration, and sample 64 coarse points per ray, while GNT samples 512 rays in each iteration and 192 coarse points per ray. This configuration allows us to train and evaluate MG-NeRF efficiently on only a single GeForce RTX 3090, showcasing its computational effectiveness.

## 4.2 GENERALIZATION TO UNSEEN SCENES

MG-NeRF is evaluated against six prominent methods for generalizable NeRF: pixelNeRF (Yu et al. (2021b)), MVSNeRF (Chen et al. (2021)), IBRNet (Wang et al. (2021)), NeuRay (Liu et al. (2022)), GPNR (Suhail et al. (2022)), and GNT (Varma et al. (2022)). The evaluation is conducted on the LLFF dataset (Mildenhall et al. (2019)) and the NeRF Synthetic dataset (Mildenhall et al. (2020)). Quantitative results are presented in Tab. 1, while qualitative results are illustrated in Fig. 7. Although MG-NeRF achieves a slightly lower score compared to GNT, the current state-of-the-art method, it excels in synthesizing novel views just as effectively. Futher, MG-NeRF performs approximately 51.7% worse than GNT on the NeRF Synthetic dataset while approximately 40.4% worse on the LLFF dataset. This discrepancy can be attributed to the complexity of the LLFF dataset (Mildenhall et al. (2019)), which contains more intricate semantic information compared to the synthetic dataset, where the semantic information of out MG-NeRF could demonstrate its influence better.

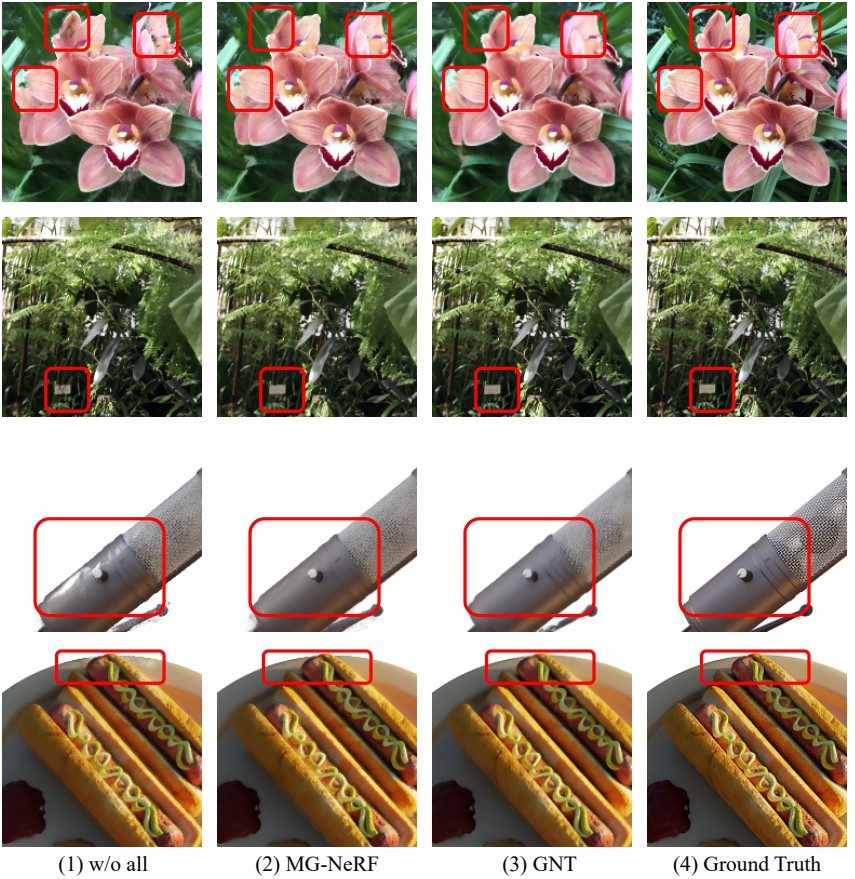

| (1) w/o all | (2) MG-NeRF | (3) GNT | (4) Ground Truth |

Figure 7: Experiments all-in-one results. (1) Without the proposed methods, artifacts are generated apparently. (2)(3)(4) MG-NeRF is able to synthesize high quality novel views the same as GNT (SOTA).

### 4.3 Ablations and Analysis

We conducted five separate experiments to evaluate the effectiveness of the proposed four methods. The quantitative and qualitative results are presented in Tab. 2 and Fig. 7, respectively. Each of the four methods demonstrates significant improvements compared to the baseline experiment (w/o all). In the absence of our methods, noticeable artifacts are observed in the synthesized novel views, as depicted in Fig. 7(1). Besides, we observed that the frequency regularization (Mask) technique notably enhances NeRF's generalization ability, particularly on the NeRF Synthetic dataset (Mildenhall et al. (2020)). This phenomena can be attributed to the fact that the NeRF Synthetic dataset (Mildenhall et al. (2020)) contains images with transparent backgrounds, which tend to exhibit more high-frequency information.

## 5 Conclusion

We have introduced MG-NeRF, a pioneering multimodal representation learning framework designed to enhance the generalization capabilities of NeRF. In this framework, we generate text modality data for the current mainstream image datasets commonly used in NeRF's generalization process. MG-NeRF leverages enriched features with semantic supervision and frequency regularization to learn multimodal knowledge. Experimental results demonstrate that MG-NeRF significantly improves the generalization ability of NeRF. There is still potential for further enhancement by incorporating additional modalities and resources. We anticipate that our work will inspire and guide future research endeavors in this domain.

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
