# OpenReview forum: "MG-NeRF: Multimodal Representation Learning for Generalizable NeRF"
_ICLR.cc/2024/Conference — ICLR 2024 Conference Withdrawn Submission_

### Official Review · Reviewer_5MMB · 2023-10-28

**Soundness:** 1 poor
**Presentation:** 3 good
**Contribution:** 1 poor
**Rating:** 1
**Confidence:** 4

**Summary:**

(I'm sorry that, in the initial version of my post, there was a misinterpretation of the paper. Therefore, I have posted a revised review.")

The paper tackles novel view synthesis with GNT-based neural radiance fields and proposes (1) additional ray features by pretrained vision-language models, (2) semantic loss, and (3) frequency regularization.
The feature is a combination of a text feature, whose text is based on a pretrained caption generator, and an image (pixel-level) feature, encoded by IMAGEBIND.
Semantic loss computes cosine similarity between a re-encoded feature of a rendering image and its text feature by IMAGEBIND.
Frequency regularization is similar to existing methods by Yang et al. (2023) and gradually trains low to high-frequency positional features.

**Strengths:**

- Attempt to use additional pretrained model-based features for GNT-based NVS.
- Demonstrate the effect of semantic loss using auto-generated captions

**Weaknesses:**

- The effects and claims are not well validated. It is very difficult to interpret the paper's contributions.
  - In my understanding, the proposed method is an extension of GNT. However, the performance of the proposed method (MG-NeRF) is much worse than GNT (and other baselines) in all the datasets in Table 1. Although ablation tests in Table 2 could show the effect of the proposed techniques, they are not so comprehensive, sound, and fair for claims.
  - For example, GNT's image encoder employs ResNet34, and IMAGEBIND (used for MG-NeRF) employs a much larger ViT. The image encoder architectures for extracting ray features differ. Thus, Fusion's improvement could be caused by just the performance of an image encoder architecture rather than multi-modality or even pre-training at worst.
  - Various semantic-aware loss functions have already been proposed in prior work. If the paper introduces a variant for a contribution, it requires comparisons. Such loss functions include SSIM, VGG perceptual loss, and CLIP semantic consistency (Jain et al. 2021).
  - If the paper wants to claim "MG-NeRF achieves appreciable achievement ... even trained with considerably less resources than prior work," experiments for the demonstration and discussion are required but not found.
- Less surprise. It is not so surprising that the performance is improved when using some different encoder pretrained on (additional) large-scale datasets. If the paper could provide insight into one more thing (e.g., why language/multi-modality is important, better captioning criteria, etc.), the contribution would be improved.
- The quality of the manuscript is not great. The experiment does not discuss the difference between the methods in detail. Related work is not well connected to this work. Figure 1 is plagiarized from the paper by Mildenhall et al. (2020). Overall, the claims are unclear and less convincing.

**Questions:**

- If the MG-NeRF is based on GNT, the performance of "w/o all" in Table 2 could be matched with "GNT" in Table 1. Why did they differ? Did it stem from the number of rendering samples (64 v.s. 192 in Training / Inference Details)? Fair and appropriate comparisons would make the contributions clearer.
- In Table 2, does "w/ LoRA" mean "w/o all + LoRA" or "w/o all + Fusion + LoRA"? If the former, I cannot understand the model (because LoRA should be used for Fusion). If the latter, I wonder what is different between "w/ Lsec" (i.e., w/o all + Fusion + LoRA + Mask + Lsec) and "MG-NeRF," and whether full-finetuning (instead of LoRA) improves MG-NeRF or not.
- Please proofread and carefully write the paper. As a simple example, the usage of "neural radiation field" (on page 3) is terrible as a NeRF paper. The current manuscript still has ample room for improvement. After addressing these issues, the true value of your research will be effectively communicated to the readers.
- Missing citations. For example, please add discussion to these papers related to ideas using captions for improving NeRF.
  - NeRDi: Single-View NeRF Synthesis with Language-Guided Diffusion as General Image Priors, Deng et al., 2022
  - Deceptive-NeRF: Enhancing NeRF Reconstruction using Pseudo-Observations from Diffusion Models, Lie et al., 2023

---

### Official Review · Reviewer_dNY8 · 2023-10-30

**Soundness:** 2 fair
**Presentation:** 3 good
**Contribution:** 2 fair
**Rating:** 5
**Confidence:** 4

**Summary:**

This paper aims to employ a vision language pre-training framework for NeRF’s generation to synthesize novel views of unseen scenes. To this end, the authors employ a vision language pre-training (VLP) framework to generate text information per scene and then fuse and feed the image and text features to NeRF. Subsequently, a semantic loss could be introduced to encourage NeRF to synthesize reasonable novel views. Besides, a frequency regularization mechanism for positional encoding is introduced to prevent NeRF from overfitting to high-frequency information. The numerical experiment and visualizations show the acceptable performance of the proposed method on two evaluation datasets.

**Strengths:**

•	The paper is easy to follow, with a clear method description and corresponding figures.

**Weaknesses:**

* The degraded reconstruction performance. The proposed MG-NeRF aims to employ a vision language pretraining framework, i.e., ImageBind, to increase the NeRF’s generalization on synthesizing novel views of unseen scenes. Since the two-stage transformer-based NeRF network adopted MG-NeRF is similar to GNT (Varma et al., 2022), the performance of MG-NeRF is expected to approach or surpass GNT. However, the experiments show a negative effect while employing MG-NeRF.

* For the goal of synthesizing novel views of unseen scenes, the conducted experiments could not support this goal. It needs to be clarified what the benefits of MG-NeRF are while dealing with unseen scenes. Besides, the role of the language input should be discussed in conducting related experiments to demonstrate its advantages.

**Questions:**

The primary concern of this paper is weak experiments. The current experiments could not sufficiently support the advantage of using multimodality via the proposed MG-NeRF.

---

### Official Review · Reviewer_diGW · 2023-11-01

**Soundness:** 3 good
**Presentation:** 4 excellent
**Contribution:** 2 fair
**Rating:** 3
**Confidence:** 4

**Summary:**

The authors present MG-NeRF, a generalizable NeRF framework that takes multi-modal information into consideration for the NeRF scene representation. Specifically, the authors adopt language as the other modality beyond the image modality to enable image-text cross-modality learning when rendering novel views using generalizable NeRF. The authors use pretrained vision-language models to obtain aligned image and text features that can be used for NeRF rendering. The authors also adopt additional techniques like LoRA finetuning, semantic loss, frequency regularization to obtain better performance. Experimental results show that MG-NeRF outperforms part of the baselines and the techniques applied in MG-NeRF like LoRA, semantic loss can contribute to better performance.

**Strengths:**

++ To my knowledge, this is the first work to consider incorporating the text modality into learning NeRF representation. This idea is novel and deserves to be dived into, and this paper could be a pioneering work in this direction.

++ Given the current rapid development of vision-language models, investigating vision-language models for NeRF representation could be a promising direction for future research.

++ I find that it is a good way to obtain joint embedding from both the image and text modalities from a pretrained vision-language model. The information obtained in this way might be more informative than using discriminative modules like ResNet or Transformer blocks which previous conditional NeRF methods typically use.

++ The presentation of this paper is clear and easy to follow. The authors explain in details and thoroughly about how the proposed MG-NeRF works.

**Weaknesses:**

-- I think the most crucial weakness of this paper is that the performance of the proposed MG-NeRF is actually worse than many compared methods like IBRNet, NeuRay, GPNR and GNT, especially as for IBRNet it is a CVPR 2021 paper so it is a relatively old baseline. The authors claim in the paper that "MG-NeRF achieves a slightly lower score". However, I think more than 2 dB worse than GNT and more than 3 dB worse than NeuRay in PSNR evaluated on NeRF Synthetic dataset are already significantly inferior performances. I acknowledge the novel idea proposed by authors of incorporating text modality into NeRF. However, the inferior performance is kind of severe from my point of view. The authors are encouraged to explain clearly for this point in order to raise my score. I also have additional comments regarding this point in the "Questions" part of my review.

-- It might be a little overclaiming to say that this work is the first to consider multi-modal information in NeRF. I think it is indeed the first one to consider the text modality. However, for other modalities, as far as I know, [1] considers the acoustic information, while [2] considers other modalities like point cloud and infrared images.

-- The authors mention that MG-NeRF is more computationally efficient so that it can be trained on only a single GeForce RTX 3090 GPU. Could the authors also show the computational cost of other conditional NeRF models so that we could have a more complete comparison?

[1] Luo et al. Learning Neural Acoustic Fields. NeurIPS 2022.

[2] Zhu et al. Multimodal Neural Radiance Field. ICRA 2023.

**Questions:**

-- This is related to my concern on the MG-NeRF performance in the "Weaknesses" part. In the ablation, the performance of the "w/o all" variant of MG-NeRF is relatively low compared with other baselines (especially on the NeRF Synthetic dataset). In principle, since MG-NeRF can be a general purpose model design, I think it can be instantiated on different conditional NeRF backbones like the more powerful GNT or NeuRay. Thus, I am expecting that the "w/o all" variant of MG-NeRF should reach at least comparable results as the other conditional NeRF models. I am wondering whether the authors have tried building MG-NeRF upon other conditional NeRF backbones and in that case, whether the performance of the full MG-NeRF could be better than the results shown in the current manuscript.

-- For the frequency regularization part, I feel like it is a general purpose design and can be applied on any conditional NeRF models, but relatively loosely related to the motivation of proposing MG-NeRF (incorporating the text modality into NeRF). With that being said, I am wondering whether the frequency regularization could provide benefits. Also, are there any underlying relationships of the frequency regularization technique to the text modality introduced in MG-NeRF?

---

### Official Review · Reviewer_eSq7 · 2023-11-03

**Soundness:** 2 fair
**Presentation:** 2 fair
**Contribution:** 1 poor
**Rating:** 1
**Confidence:** 4

**Summary:**

The authors propose MG-NeRF, a framework that incorporates multi-modal information to improve NeRF's generalization. The authors first propose a fully automatic strategy to derive a text description of a scene from multi-view images. The network is optimized jointly for reconstruction along with a semantic loss that measures the similarity between the rendered view and the input text description. The authors also propose a strategy to regularise NeRF training to avoid overfitting to the high-frequency input components. Results indicate that the method performs reasonably compared to other SOTA methods.

**Strengths:**

- The paper is reasonably easy to follow.
- Sufficient ablations are conducted and the authors have made sure to compare against recent SOTA generalization methods.

**Weaknesses:**

- The motivation to use multi-modal input(s) is not really clear. For example, one can perhaps think that regularising the rendered multi-view images to be consistent with a single text prompt can perhaps improve consistency or can provide semantic awareness to NeRFs intermediate that can perhaps improve rendering quality. (nevertheless, the above reasons do not concretely motivate the choice of using multi-modal inputs)
- The results of the proposed method are quite significantly worse than other baselines. Ignoring SOTA, performance on both NeRF LLFF and NeRF Synthetic is worse than IBRNet (quite an old work).

Overall, I think the work at its current stage has multiple concerns - lack of clear motivation, poor performance, etc and is not ready for ICLR.

**Questions:**

- I urge the authors to clearly highlight the motivation to use multi-modal inputs. Perhaps this extra regularization can enable, for example, few-shot rendering. Since we do provide text embedding as input, can this be used to enable the NeRF's intermediate to be more semantic aware, etc?